# A computational platform for high-throughput analysis of RNA sequences and modifications by mass spectrometry

Samuel Wein[1,2,9], Byron Andrews[3,9], Timo Sachsenberg[4], Helena Santos-Rosa[5], Oliver Kohlbacher [2,4,6,7,8], Tony Kouzarides [5], Benjamin A. Garcia[1✉] & Hendrik Weisser [3✉]

The field of epitranscriptomics continues to reveal how post-transcriptional modification of RNA affects a wide variety of biological phenomena. A pivotal challenge in this area is the identification of modified RNA residues within their sequence contexts. Mass spectrometry (MS) offers a comprehensive solution by using analogous approaches to shotgun proteomics. However, software support for the analysis of RNA MS data is inadequate at present and does not allow high-throughput processing. Existing software solutions lack the raw performance and statistical grounding to efficiently handle the numerous modifications found on RNA. We present a free and open-source database search engine for RNA MS data, called NucleicAcidSearchEngine (NASE), that addresses these shortcomings. We demonstrate the capability of NASE to reliably identify a wide range of modified RNA sequences in four original datasets of varying complexity. In human tRNA, we characterize over 20 different modification types simultaneously and find many cases of incomplete modification.

[1] Epigenetics Program, Perelman School of Medicine, University of Pennsylvania, Philadelphia, PA, USA. [2] Center for Bioinformatics Tübingen, University of Tübingen, Tübingen, Germany. [3] STORM Therapeutics Limited, Moneta Building, Babraham Research Campus, Cambridge, UK. [4] Applied Bioinformatics, Department for Computer Science, University of Tübingen, Tübingen, Germany. [5] Gurdon Institute, University of Cambridge, Cambridge, UK. [6] Quantitative Biology Center, University of Tübingen, Tübingen, Germany. [7] Biomolecular Interactions, Max Planck Institute for Developmental Biology, Tübingen, Germany. [8] Translational Bioinformatics, University Hospital Tübingen, Tübingen, Germany. [9] These authors contributed equally: Samuel Wein, Byron Andrews. ✉email: bgarci@pennmedicine.upenn.edu; hendrik.weisser@stormtherapeutics.com

RNA is an extensively modified biological macromolecule. Over 150 chemically distinct modifications have been reported. The presence of methylated adenine, cytosine, and guanine in RNA was uncovered in the 1960s[1], and pseudouridine has been referred to as the "fifth base" for decades[2]. However, widespread interest in these epitranscriptomic marks has been raised by recent reports that underscore their importance in a wide variety of developmental signalling. In stem cells the intracellular effector proteins SMAD2 and SMAD3 promote binding of the N6-methyladenosine (m6A) writer complex to a subset of mRNAs associated with early cell fate decisions[3]. Likewise, a number of modifications are associated with disease. It has been demonstrated that the loss of taurine modification in the anticodon of mitochondrial tRNA-Leu is responsible for mitochondrial myopathy, encephalopathy, lactic acidosis, and stroke-like episodes (MELAS)[4]. m6A is implicated in obesity[5] and associated with defects in functional axon regeneration in mice[6]. Aberrant methylation of cytosine-5 (m5C) in tRNAs has been linked to neuro-developmental disorders[7].

Recent interest in epitranscriptomics has also been spurred by technical advances in next-generation sequencing (NGS) technology, which has allowed modifications in mRNA to be profiled individually. All of the approaches based on Solexa/Illumina sequencing use antibodies to immunoprecipitate modified RNA, and/or apply chemical or enzymatic treatments to alter it and read out modifications as mutations or truncations in the preparation of cDNA[8,9]. The primary caveat of these methods is that only a single type of modification can be profiled in each experiment, and specific chemical, enzymatic and/or antibody reagents do not exist for every modification. Further complications can be caused by lack of specificity of the existing antibodies, in particular m6A and m6Am[10]. Steps have been made towards uncovering modifications directly using long-read sequencing platforms[11,12], but many technical challenges stand between these approaches and routine use, not least a significant error rate in base calling[13]. NGS-based methods have also generated conflicting results in the past[14,15], underscoring the need for orthogonal approaches.

Mass spectrometry (MS) is currently the only technique that can directly and comprehensively characterize chemical modifications in RNA sequences. The majority of RNA MS has focused on reducing the RNA to mono-nucleosides and applying workflows analogous to metabolite analysis[16]. While these techniques are effective in determining what modifications are present in a sample, all information about the location and co-occurrence of modifications is lost. This information is critical in complex samples to attribute modifications to specific RNAs. Even in simpler cases, modification location and co-occurrence may be important for a phenotypic effect; for example, in microRNA 2′-O-methylation of the 3′-most nucleic acid sterically inhibits 3′ exonuclease digestion, which increases the half-life of the modified microRNA in the cell[17]. For this reason there is an interest in analyzing samples in as close to their native states as possible. Analysis of intact RNA oligonucleotides by tandem mass spectrometry (MS/MS) is capable of determining modification sites with single-nucleotide resolution, by comparing mass spectra with a sequence database[18]. However, oligonucleotides are challenging to separate via mass spectrometry-compatible liquid chromatography (LC). The current approach of choice is reversed-phase ion-pair liquid chromatography[19].

In addition to the experimental challenges, difficulties emerge in interpreting the acquired data. Considerable efforts toward automating data analysis have been made in recent years, starting with SOS[20] in 2002, Ariadne[21] in 2009, OMA/OPA[22] in 2012, and RNAModMapper[23] in 2017, all of which are programs for database-matching or decoding the complicated patterns of oligonucleotide fragmentation. However, none of the existing software solutions offers key features necessary to analyze data from large-scale experiments. First, no software can efficiently handle the analysis of RNA oligonucleotide data—especially of more complex samples or involving many different modifications—in batch-compatible fashion. Second, statistical validation strategies, such as false-discovery rate (FDR) estimation, are not implemented. This leads to unreliable sequence assignments and subjective manual assessment of spectra for validation. Third, existing solutions do not tie into any larger analytical framework, making integration with other (e.g. quantitative) data difficult. In contrast, shotgun proteomics has been sequencing peptides reliably for many years, and the inference, identification and quantification of proteins from constituent peptides has been automated to such a degree that the field has matured into answering biological questions at a more fundamental level[24].

To fill this fundamental gap, we present a fast, scalable database-matching tool called NucleicAcidSearchEngine (NASE) for the identification of RNA oligonucleotide tandem mass spectra. Our software is implemented within the OpenMS framework, an open-source toolset for processing mass spectrometric data[25]. NASE will be fully integrated into the primary distribution of OpenMS in the upcoming version 2.5, and will then be available for download as part of OpenMS at https://www.openms.de. In the meantime OpenMS builds containing NASE are available at https://www.openms.de/nase. Beyond speed and sensitivity, NASE provides advanced features like FDR estimation, precursor mass correction, and support for salt adducts. Powerful visualization capabilities are available through OpenMS' data viewer. By supporting the common interface of The OpenMS Proteomics Pipeline[26], NASE can be easily used in automated data analysis workflows. This interoperability also enables the label-free quantification of RNA oligonucleotides based on NASE search results.

Using four original datasets we demonstrate the capability of NASE to reliably identify a variety of RNA types from different sources, and show how data visualization and label-free quantification can augment the interpretation of identification results.

## Results

**RNA oligonucleotide MS datasets.** Using nanoflow ion-pair liquid chromatography coupled to high-resolution tandem mass spectrometry (nLC-MS/MS), we generated four datasets from RNA samples of increasing complexity. First, oligonucleotides with the sequence of mature Drosophila let-7 microRNA, 21 nt in length, were produced synthetically in unmodified and modified (2′-O-methylated at the 3′ uridine) forms ("synthetic miRNA" dataset). We characterized a 1:1 mixture of both forms of this RNA. Replicate measurements were acquired using different normalized collision energy (NCE) settings in the mass spectrometer. Second, we prepared two samples of an in vitro-transcribed yeast lncRNA (NME1, 340 nt long), one of which was treated with an RNA methyltransferase (NCL1) catalyzing the 5-methylcytidine (m5C) modification ("NME1" dataset). Third, we used size exclusion chromatography to produce two samples containing long ribosomal RNAs (18S and 28S) from a human cell line ("human rRNA" dataset). Fourth, we generated three biological replicates of human total tRNA from a cellular extract — a complex mixture of highly modified RNAs ("human tRNA" dataset). The "NME1", "human rRNA" and "human tRNA" samples were all digested with an RNA endonuclease (RNase T1) to generate oligonucleotide sequences of a length amenable to mass spectrometry.

**A powerful search engine for RNA MS data.** We developed a sequence database search engine for the identification of (modified) RNA sequences based on tandem mass spectra. The software, termed NucleicAcidSearchEngine (NASE), was implemented within the OpenMS framework and combines existing functionality (e.g. for data input/output, filtering, and FDR estimation) with newly developed features. (See Methods section for details.) Given a mass spectrometry data file and a FASTA file containing target and decoy (shuffled or reversed) RNA sequences as inputs, NASE generates oligonucleotide-spectrum matches with statistically meaningful FDR scores. OpenMS' interactive viewer, TOPPView[27], was extended to support RNA identification results obtained using NASE, mirroring and augmenting existing functionality for visualizing peptide identifications in proteomics experiments.

In addition to the built-in FDR calculation, NASE provides other features that set it apart from alternative tools that are currently available. Even with extensive preparation, nucleotide samples frequently contain salt adducts (in the form of cations attached to the phosphate backbone). NASE searches can take this into account, by allowing users to specify chemical formulas of adducts to consider in the precursor mass comparisons.

Furthermore, NASE supports the correction of precursor masses for MS2 spectra that were sampled from isotopologue peaks other than the monoisotopic one. Especially for longer sequences, MS2 precursor ions are often picked from higher-intensity, heavier isotopologues by the mass spectrometer's data-dependent acquisition software. Without adjustment, the precursor masses would not closely match the theoretical (monoisotopic) masses of the correct oligonucleotides, leading to no assignment or incorrect matches. We implemented a correction that considers offsets corresponding to multiples of a neutron mass when comparing precursor and oligonucleotide masses. This feature greatly increases NASE's ability to identify oligonucleotides with longer sequences. Curiously, we observed cases where the instrument software erroneously estimated the precursor ("selected ion") m/z value to be below the apparent monoisotopic peak. We found that this could be corrected by allowing a negative offset (−1) in the precursor mass correction.

Finally, through the OpenMS toolbox NASE enables seamless label-free quantification of the oligonucleotides that were identified in a sample. A corresponding analysis pipeline can be easily created and run using a graphical workflow editor. Supplementary Fig. 1 shows an example pipeline from our analysis of the NME1 data, using the editor that is conveniently included with OpenMS[28].

**MS-based sequencing of an intact synthetic microRNA.** In our analysis of data from the synthetic miRNA sample, we found a strong dependence of sequence coverage on the Normalized Collision Energy (NCE) value. Identical samples were run with NCE ranging from 5 to 55. The best results were obtained for an NCE of 20 (Supplementary Fig. 2). Subsequent LC-MS/MS analyses, including of the NME1 and tRNA samples, were thus carried out with this NCE setting.

At the optimal NCE, both unmodified and modified RNA were detected, and the location of the modification could be determined with high confidence. 874 spectra were identified that passed our hyperscore cutoff, matching sequences of length 5–21 nt, including the full-length let-7. The shorter sequences correspond to artefacts of incomplete solid-phase RNA synthesis, which are easily detectable by LC-MS. In the full 21-nt sequence we averaged over two-fold MS2 ion coverage of the let-7 sequence, with one or more forward (a-B/a/b/c/d) ion and one or more reverse (w/x/y/z) ion detected at each base (see Fig. 1, ion naming scheme from McLuckey et al.[29]). This demonstrates our ability to sequence even relatively long (>20 nt) RNAs.

**Performance comparison of search engines for RNA MS data.** We processed the NME1 data using the three search engines Ariadne, RNAModMapper, and NASE. We ran target/decoy database searches using m5C as a variable modification and combined results from all replicates. We then compared the search engines in terms of: A, the number of identified spectra at different FDR thresholds; B, the sequence length distribution of the identified oligonucleotides at 5% FDR (Fig. 2). NASE identified significantly more spectra at a given confidence level than the other tools. It also found longer oligonucleotides, which would be more informative for identifying RNAs in complex samples. About 10% of the oligonucleotide-spectrum matches generated by NASE at 1% FDR included sodium (2.9%) or

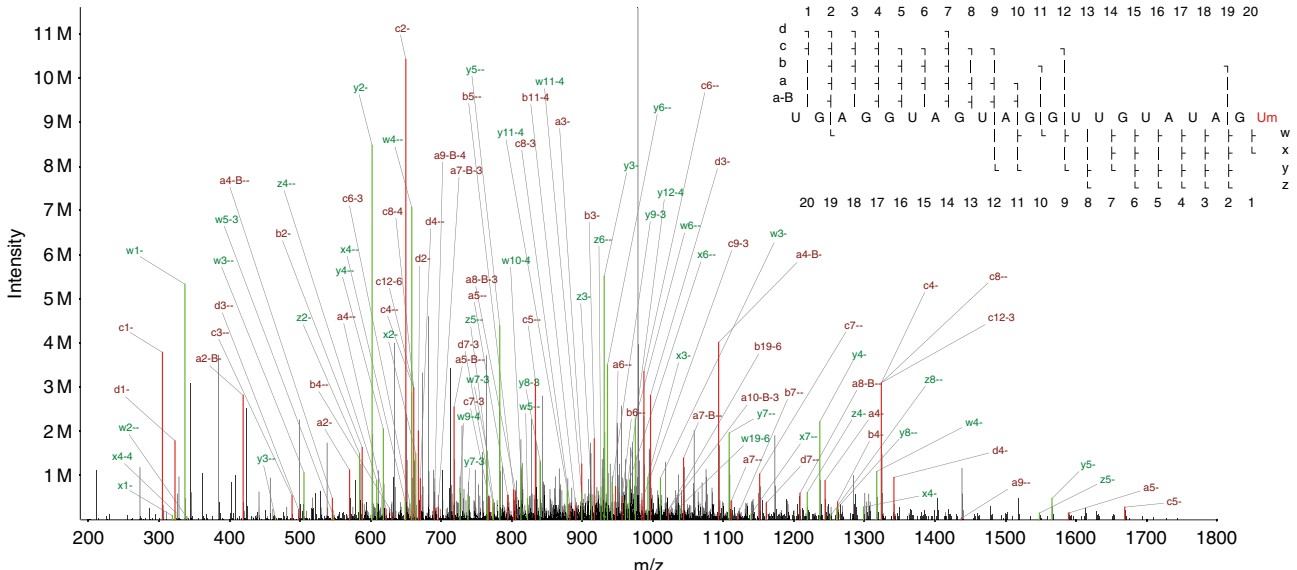

**Fig. 1 A tandem mass spectrum of synthetic let-7 denoting all of the assigned peaks.** a-B/a/b/c/d ions are shown in red, w/x/y/z ions in green. The primary ion was deprotonated seven times to give a charge state of −7 (m/z 971.55). The ion coverage plot in the upper right shows coverage for nine different types of fragment ion.

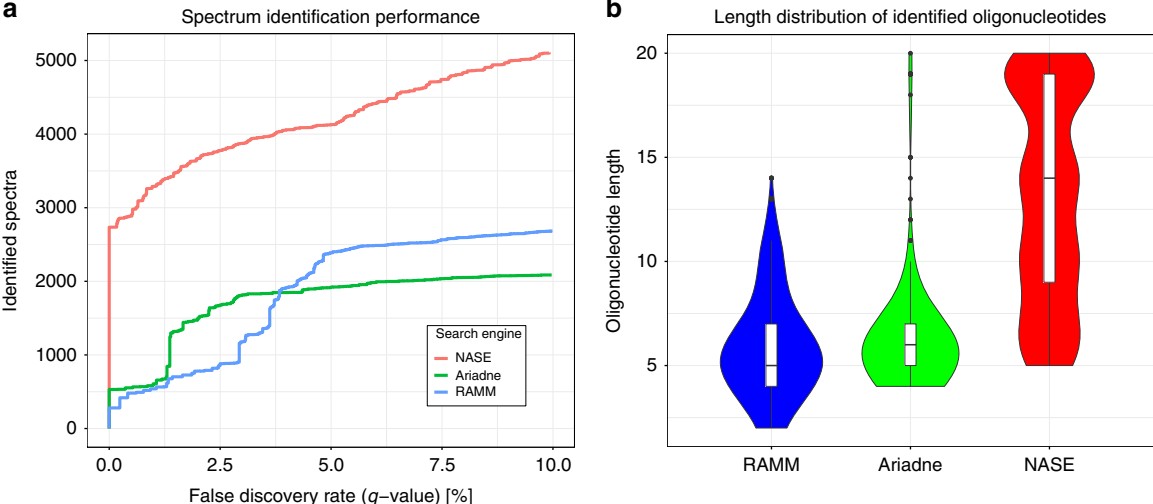

**Fig. 2 Performance comparison of RNA identification engines—Ariadne, RNAModMapper (RAMM) and NucleicAcidSearchEngine (NASE)—based on searches of the NME1 data. a** Number of successfully identified spectra plotted against the q-value, a measure of the false-discovery rate, which was calculated from a target/decoy database search using each of the three tools. **b** Sequence length distribution of identified oligonucleotides for each tool at a confidence level of 5% FDR, shown using violin and box plots. Box plot center lines indicate data medians, box limits show upper and lower quartiles, whiskers extend to the largest/smallest value no further than 1.5 times the interquartile range from the box, black dots represent outliers.

potassium (7.2%) adducts and would have been missed without the adduct search capabilities.

Note that Ariadne's performance in this comparison was hampered by the fact that a recommended tool for data preprocessing, the commercial software SpiceCmd, was not available to us. RNAModMapper had previously been evaluated based on searches against "expected" sequences only (i.e. no decoys), followed by manual validation of spectral assignments[23,30].

**Detection of differential methylation sites**. To assess the performance of our software at detecting RNA modifications, we compared the NASE search results for the NME1 lncRNA with and without NCL1 incubation (Fig. 3a). Following common practice in the proteomics field[31], we considered results at a high confidence level after filtering to 1% FDR and removing "single hits" (oligonucleotides identified only based on a single spectrum) in each run. At this level, 72% and 73% sequence coverage were achieved for the control and the NCL1-treated sample, respectively. As Fig. 3a shows, there is good agreement between the unmodified oligonucleotides that were identified in both samples, indicating that our method works reproducibly. In the high-confidence set, m5C-modified oligonucleotides were only found in the NCL1-treated sample where they would be expected. Two illustrative examples are the isobaric oligonucleotides "UCA-CAAAU[m5C]G" (at position 21-30 in the NME1 sequence) and "UAAC[m5C]CAAUG" (pos. 299-308) that were identified based on 29 and 10 spectra in multiple charge states ($-2$ to $-5$). Two additional identifications were made of the sequence "UAACC [m5C]AAUG", i.e. with a shifted localization of the modification. Figure 4a shows a corresponding data section from the NCL1-treated sample, visualized as a two-dimensional LC-MS map. Identifications of the unmodified, adducted, and modified variants of the two oligonucleotides are displayed in the context of MS1 signal intensities. At the bottom, "UCACAAAUCGp" (left) and "UAACCCAAUGp" (right) can be seen eluting in overlapping peaks. (In our notation, "p" at the end of a sequence represents the 3′ phosphate generated by RNase T1 cleavage.) In the middle, the corresponding mono-methylated oligonucleotides are convincingly detected, with a mass shift of 14 Da and a slight RT shift relative to their unmethylated counterparts. At the top,

the unmodified oligonucleotides were identified with a sodium adduct (mass shift of 22 Da). A corresponding image showing the loss of signal for the modified oligonucleotides in the control sample is available as Supplementary Fig. 3. In Fig. 4b we compare spectrum matches for the two modified oligonucleotides, showcasing the high quality of the matches as well as our MS2 visualization capabilities, including the newly added ion coverage diagrams.

**Label-free quantification of RNA MS data**. We quantified the identified oligonucleotides in the two NME1 samples, using a label-free, feature detection-based approach. Figure 3b summarizes the results. Although all oligonucleotides come from the same RNA, they were quantified with signal intensities spanning several orders of magnitude. This is indicative of widely varying ionization efficiencies during MS analysis, a common caveat that generally limits label-free quantification to relative comparisons between similar samples.

Of 26 and 36 different oligonucleotide sequences that were identified as part of the high-confidence set in the control and NCL1-treated sample, respectively, 25 and 32 could be quantified in either sample based on at least one replicate (corresponding to 96% and 89% success rates). Unmodified oligonucleotides were quantified at similar levels in both NME1 samples, with a correlation of 0.94 of the log-intensities per sample (median of the technical replicates) for oligonucleotides in all charge/adduct variants (0.98 when considering only the "best" variant in terms of reproducibility across replicates, as measured by the coefficient of variation). Methylated oligonucleotides were only identified and quantified in the NCL1-treated sample, but their unmodified counterparts exhibited lower feature intensities in the treated sample compared to the control, consistent with a partial shift of the ion current to the modified variants (Supplementary Fig. 4).

More advanced capabilities for LC-MS-based quantification, including retention time alignment, inference of identified analytes across samples, and labelling approaches, are already available in OpenMS for proteomics experiments. With future improvements to the support for nucleic acids in the framework, these features will become available for RNA analyses as well.

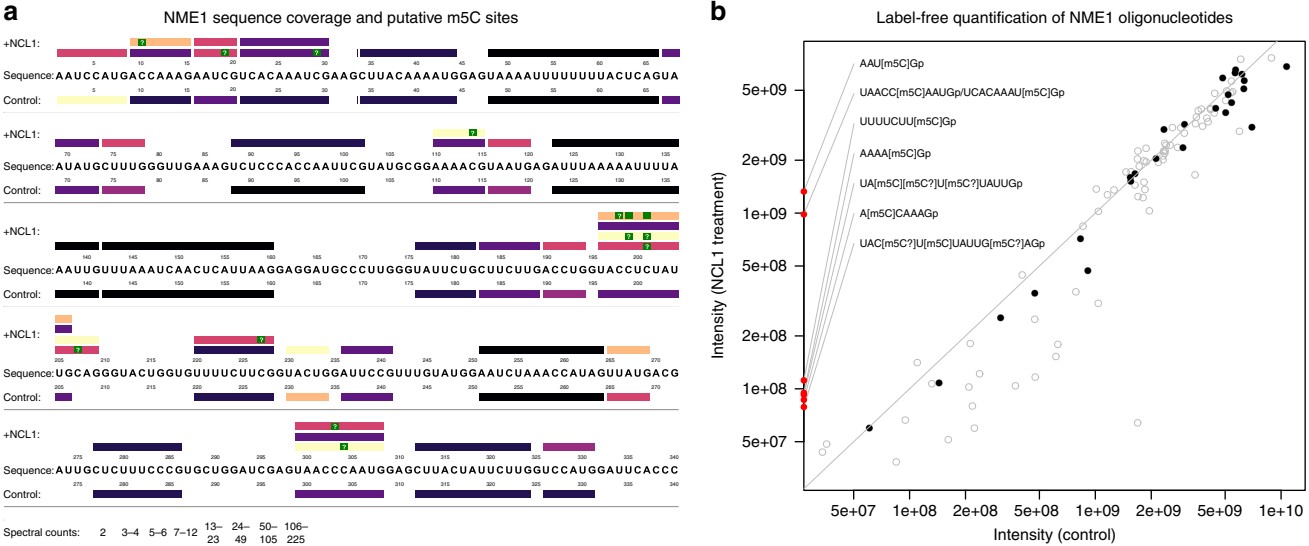

**Fig. 3 Comparison of the NME1 control and NCL1-treated sample based on NASE search results at 1% FDR, without "single hits". a** Coverage plot showing oligonucleotides identified in the respective sample above/below the NME1 RNA sequence. Bars representing oligonucleotides are colored according to their number of spectral matches (spectral counts). Oligonucleotide identifications supported by higher spectral counts (darker colors) are considered more reliable. Putative 5-methylcytidine (m5C) modification sites are marked in green. Sites with a "?", the one-letter code for m5C, were uniquely localized; "blank" sites indicate uncertainty between two possible locations, due to the absence of discriminating peaks in the corresponding mass spectrum. **b** Label-free quantification results for identified oligonucleotides, comparing feature-based signal intensities in the two samples (averaged across replicates). Gray circles show all individual charge/adduct states, while black dots indicate the "best" representative (lowest coefficient of variation across replicates) for quantifying each oligonucleotide. Oligonucleotides that were only identified and quantified in the NCL1-treated sample, all of them methylated, are depicted on the *y* axis. The notation "[m5C?]" is used here for cases where the methylation could not be uniquely localized. No modified oligonucleotides were identified in the control sample. The gray diagonal line represents equal intensity in both samples.

**Comparative analysis of human ribosomal RNA.** Taoka et al. recently published a complete landscape of the modifications on human ribosomal RNA in TK6 cells[32]. We acquired LC-MS/MS data on human rRNA in HAP1 cells, analyzed it using NASE and compared our results to their published findings (see Supplementary Data 1). For this comparison we filtered spectrum matches to 5% FDR and removed "single hits"; where oligonucleotides with the same raw sequence were identified in different modification states (incl. unmodified), only the one with highest spectral count was considered. Our approach achieved 39% sequence coverage for 18S rRNA and 25% for 28S rRNA, which is far from complete, but much higher than the coverage achieved by Taoka et al.[32] using Ariadne on comparable data (RNase T1 digest; 13% for 18S and 7% for 28S). Using NASE we identified more and longer oligonucleotides; in the Ariadne data, almost a third of the oligonucleotides are short and match in several locations in the rRNA, making them unsuitable for modification mapping without additional information.

Taoka et al.[32] reported 68 post-transcriptional modifications (not counting pseudouridines) on 18S and 28S rRNA in regions where we have sequence coverage. For 57 of these modifications (84%) our identifications agreed exactly. In addition, five modifications were mis-localized by one position and two by two positions in our results. In three of the remaining four cases, it is plausible that the unmodified ribonucleotides were correctly detected, because the identifications are supported by high spectral counts (26, 39, 170) and the modification stoichiometry is not expected to be 100% according to Taoka et al.[32].

We further identified 10 modifications based on six oligonucleotides that are not supported by the published data, with half of these coming from only two multi-modified oligonucleotides. The oligonucleotides were all found with low spectral counts (2–4) and likely constitute false positives—with one exception,

the sequence "AUC[mG]CCCCAG" which matches in 28S rRNA and was identified 25 times. It would be an interesting candidate to investigate for cell line-specific differences in rRNA modifications.

**Analysis of a complex, highly modified transfer RNA sample.** Previous work on tRNA has shown that it is heavily modified[33]. Our analysis confirms this. We ran NASE on the "short RNA" fraction of a cell extract sample that had been digested with RNAse T1. We searched for 23 variable modifications with different molecular masses, which had previously been identified to be present in yeast or human tRNA[34,35]. Most of these represent sets of isobaric modifications which we cannot distinguish, such as position-specific variants of the same modification; e.g. "mC?" was used to represent any singly-methylated adenosine (incl. Cm, m5C etc.). Note that it was not feasible to search this dataset with this high number of variable modifications using other available database-matching tools (RNAModMapper, Ariadne).

After filtering to 5% FDR and keeping only sequences that were found in the majority of replicates, the effective FDR in the dataset was 2.7%. At this level 13,654 spectra were matched to 304 different oligonucleotides. The sequences of human tRNAs are highly similar, especially for tRNAs of one isotype, i.e. tRNAs that bind the same amino acid. Consequently, only 48 (16%) of the identified oligonucleotides map to a unique tRNA sequence; however, 234 (77%) and 289 (95%) map uniquely to a single tRNA isoacceptor (same anticodon) or isotype, respectively. Considering only the oligonucleotides that map uniquely to a specific isoacceptor, the highest sequence coverage was achieved for tRNA-Arg$^{\text{TCG}}$ (Fig. 5a). Coverage levels along the tRNA sequences were far from uniform, with the majority of identified oligonucleotides overlapping the anticodon loop and 3′ anticodon

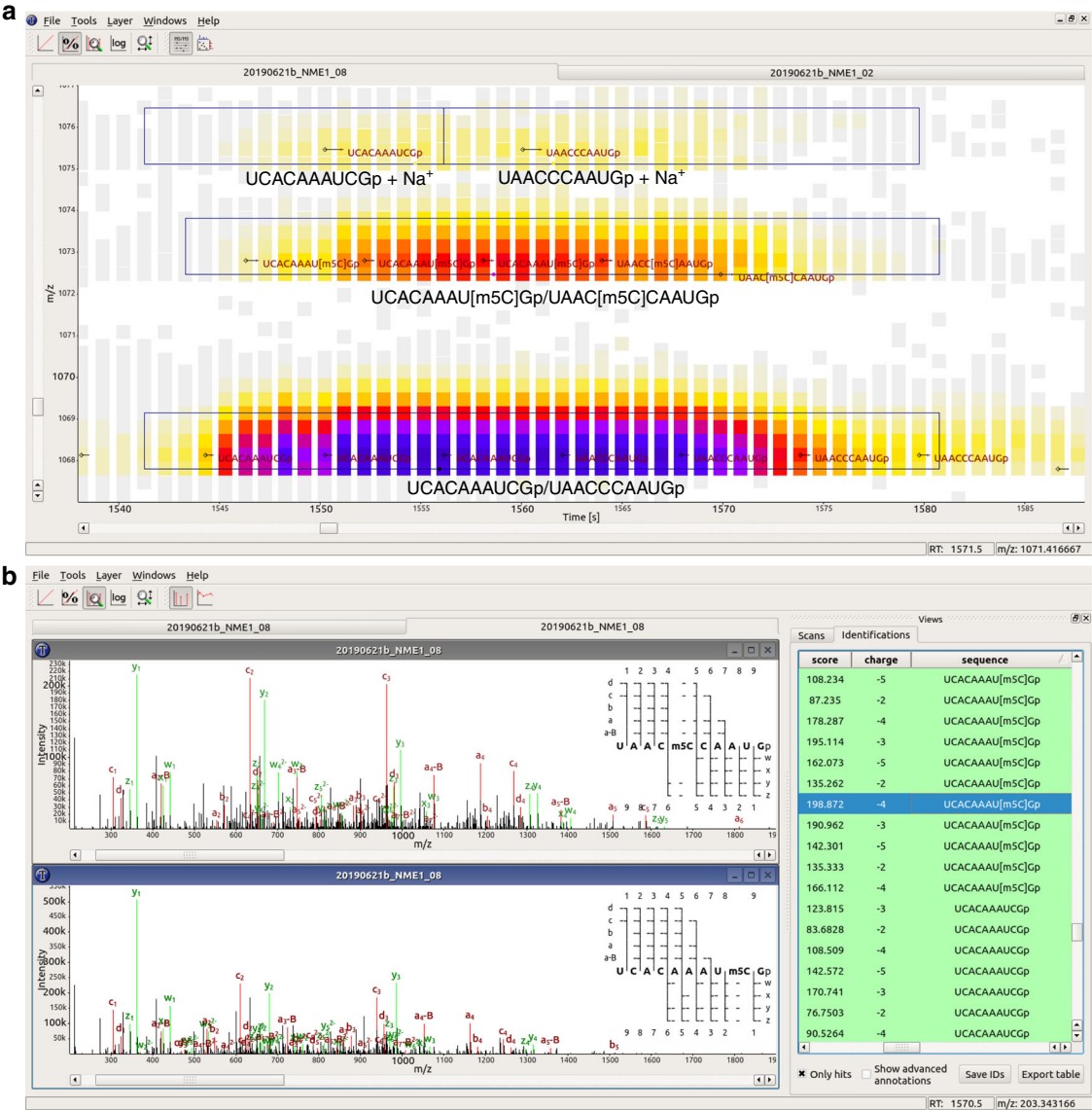

**Fig. 4 Interactive data visualization using TOPPView, showing data from the NCL1-treated NME1 sample. a** MS1 view (RT-by-m/z) of a data section. LC-MS peaks are shown as small squares, colored according to their signal intensities. Small black diamonds and horizontal lines indicate MS2 fragmentation events; oligonucleotide sequences identified by NASE from the MS2 spectra are shown in dark red font. Black boxes outline features detected for label-free quantification, which have been annotated with the corresponding oligonucleotides. All oligonucleotides shown have a charge state of −3. **b** "Identification view" comparing two MS2 spectra, identified by NASE as the sequences "UAAC[m5C]CAAUGp" and "UCACAAAU[m5C]Gp". Matching peaks between the acquired and theoretical spectra are annotated and highlighted in red and green. On the right in each spectrum plot, an ion coverage diagram shows which of the theoretical fragment ions of the sequence were matched in the MS2 spectrum (in any charge state).

stem, or the T-loop and 3′ T-stem (Fig. 5b). We hypothesize that the corresponding parts of the tRNA structure are more amenable to RNase T1 digestion than other regions.

Many of the oligonucleotides we identified contained multiple modifications. In the search, up to three modifications per oligonucleotide were allowed, to limit the combinatorial space of modified sequences that needed to be explored. Of the unique oligonucleotides identified, 12% were unmodified (accounting for 21% of the identified spectra), while 35% carried one, 28% carried two, and 25% carried three modifications (accounting for 32%, 31% and 16% of the identified spectra, respectively). All modifications considered in the search were detected as part of identified oligonucleotides. However, the prevalences of different modifications differed widely—see Table 1 for details.

Existing data on the modification landscape of human cytosolic tRNAs is incomplete (e.g. MODOMICS lists information for 36

tRNAs covering 16 isotypes) and at least some modifications are differentially regulated, complicating comparisons. We will focus on cytosine monomethylation (mC, represented by "mC?" in our search) as one example that has been studied more thoroughly, e.g. via bisulfite sequencing to detect m5C. We identified 36 unique oligonucleotides containing one (32) or two (4) mC site (s), based on a total of 1728 matched spectra. In all, 30 oligonucleotides mapped uniquely to a single tRNA isoacceptor (codon); a further four mapped to a single tRNA isotype (amino acid) but multiple isoacceptors. The two remaining oligonucleotides "AUU[mC]CAG" and "ADU[mC]CAG" could have come from either tRNA-Arg$^{ACG}$ (pos. 46-52) or tRNA-Tyr (pos. 58-64); however, for tRNA-Tyr a conserved methylation (m1A) at A58 would be expected, making tRNA-Arg$^{ACG}$ the more plausible origin. Excluding this ambiguous case, at the level of isotypes a total of 19 unique mC sites were identified. Seven of

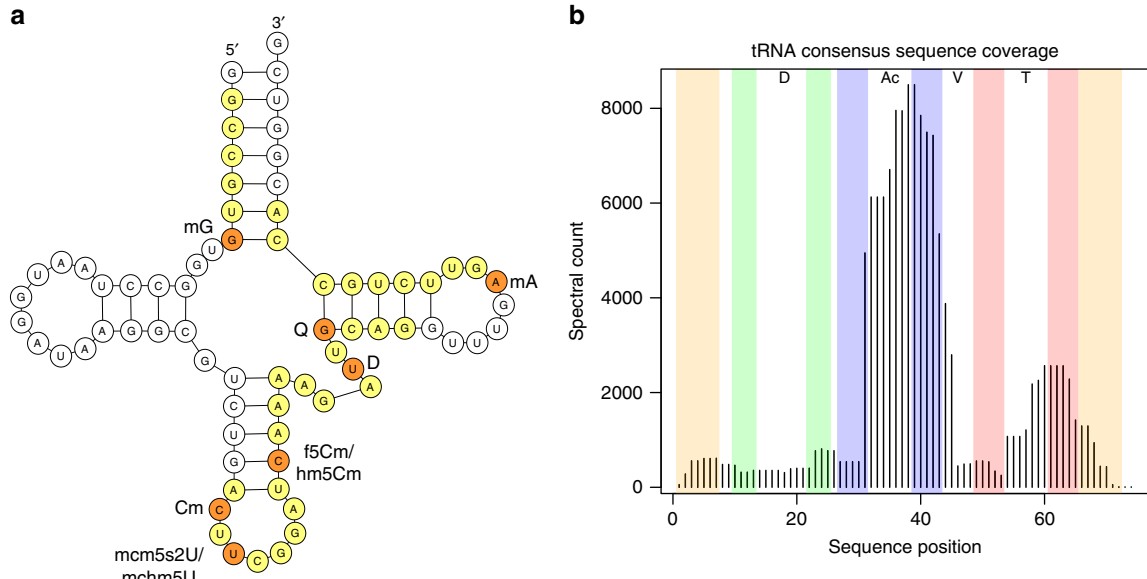

**Fig. 5 Human tRNA analysis results. a** A schematic depiction of *Homo sapiens* tRNA-Arg^TCG, showing identified sequences that map uniquely to this isoacceptor (yellow for unmodified and orange for modified residues). Total coverage is 52%. Two different modification variants were observed at U34 (wobble position) and C40. **b** Aggregated coverage of the consensus tRNA sequence by oligonucleotides identified in the human tRNA dataset. Some oligonucleotide positions in long tRNAs (tRNA-Leu, tRNA-SeC, tRNA-Ser) were adjusted to fit the consensus sequence. Complementary regions in the acceptor stem (orange), D-stem (green), anticodon stem (blue) and T-stem (red) are highlighted. D D-loop, Ac anticodon loop, V variable region, T T-loop.

| Table 1 Summary of modifications detected in the HAP1 tRNA data using NASE. | | |
|---|---|---|
| **Search mod.** | **Represented (isobaric) modification(s)** | **Spectra with mod.** |
| t6A | N6-threonylcarbamoyladenosine | 2906 |
| mA | Adenosine monomethylation (base only) | 2429 |
| mU? | Uridine or pseudouridine monomethylation (sugar or base) | 2233 |
| mC? | Cytidine monomethylation (sugar or base) | 1728 |
| mG? | Guanosine monomethylation (sugar or base) | 1512 |
| mcm5s2U | 5-methoxycarbonylmethyl-2-thiouridine | 1185 |
| D | Dihydrouridine | 998 |
| i6A | N6-isopentenyladenosine | 816 |
| m1I | 1-methylinosine | 718 |
| ac4C/f5Cm? | N4-acetylcytidine or 5-formyl-2'-O-methylcytidine | 672 |
| ms2t6A | 2-methylthio-N6-threonylcarbamoyladenosine | 633 |
| acp3U | 3-(3-amino-3-carboxypropyl)uridine or -pseudouridine | 571 |
| I | Inosine | 404 |
| m2,2G | N2,N2-dimethylguanosine | 326 |
| hm5Cm | 2'-O-methyl-5-hydroxymethylcytidine | 194 |
| Q | Queuosine | 191 |
| mchm5U | 5-(carboxyhydroxymethyl)uridine methyl ester | 187 |
| mcm5U | 5-methoxycarbonylmethyluridine | 150 |
| mcm5Um | 5-methoxycarbonylmethyl-2'-O-methyluridine | 145 |
| cm5U | 5-carboxymethyluridine | 97 |
| galQ/manQ | Galactosyl- or mannosyl-queuosine | 40 |
| yW | Wybutosine | 37 |
| hm5C | 5-hydroxymethylcytidine | 31 |

Table columns: 1. Short code of the modification specified as a search parameter. 2. The set of modifications implied by the corresponding mass shift, since e.g. positional isomers (Cm, m3C, m5C etc.) generally cannot be distinguished. A question mark in the short code (column 1) indicates that modification of the ribose or the base (implying different masses of "a-B" ions) should both be considered. Forms that are not expected to occur in human tRNA (e.g. Am, Im) are not listed. 3. Number of times identified as part of an oligonucleotide-spectrum match.

these sites agree with the "canonical" m5C sites in the VL junction of tRNAs at consensus sequence positions 48–50[7]. Other common mC sites in tRNAs are pos. 32 and pos. 34, the wobble position in the anticodon. We observed mC32 for tRNA-Arg (m3C reported in MODOMICS) and tRNA-Gln (Cm reported), and mC34 in tRNA-Met (Cm reported) and tRNA-Leu (m5C reported). In addition, we identified the oligonucleotide "A[mC] U[mC]CA[mG]" a total of 35 times, which matches at pos. 31-37

in tRNA-Trp and implies mC at both pos. 32 and 34, as well as mG at pos. 37. No data for tRNA-Trp is available in MODOMICS.

Beyond methylcytidine, known recurring modifications that we identify in several tRNAs include monomethylation at G10, dihydrouridine at U20, mono- or dimethylation at G26, N6-threonylcarbamoyladenosine (t6A) at A37 and monomethylation at A58. Interestingly, we observe 5-methoxycarbonylmethyl-2-

thiouridine (mcm5s2U) at the wobble U34 of tRNA-Arg[TCG/T], tRNA-Gln[TTG], tRNA-Glu[TTC] and tRNA-Lys[TTT]; this specific modification has been characterized in cow (tRNA-Arg[TCT]) and rat (tRNA-Glu[TTC] and tRNA-Lys[TTT]), but has not been directly located in human samples[36]. This modification, which is installed by three consecutive enzymatic steps, has been reported to be induced in oncogenic conditions, and is important for tuning the expression of protein factors based on their codon content[37].

The ability to identify and localize multiple different modifications simultaneously is an unique advantage of the oligonucleotide MS approach. In many cases we find additional, alternatively modified (or unmodified) variants of "expected" oligonucleotides. In particular, for an oligonucleotide that matches the T-loop region in several tRNA-Ala genes we robustly detect the doubly methylated form (mU55 and mA58), both singly methylated forms and the unmodified form. For the equivalent oligonucleotides in tRNA-Cys[GCA], we found at least 60 matches for each of the double methylation and a single methylation at A58. In oligonucleotides overlapping the anticodon loop and the 3′ anticodon stem, we detect multiple forms e.g. for tRNA-Glu[CTC] (unmodified and mC39), tRNA-Gly[CCC] (mU39 with and without mU32), tRNA-Met (either or both of mC34 and t6A37) and tRNA-iMet (unmodified and t6A38). In tRNA-Glu[TTC] we find mcm5s2U as well as its precursor mcm5U at U33 (presumably mis-localized from the wobble U34). For tRNA-Ser we observe several different forms in this region—primarily mA37 and t6A42 with or without mU44 for tRNA-Ser[GCT], and N6-isopentenyladenosine (i6A) at A37 with either, both or neither of mU39 and mU44 for tRNA-Ser[A/CGA]. In tRNA-Lys[TTT], among a number of identified oligonucleotides all covering pos. 31–42, the four with highest spectral counts (all above 25) show what could be interpreted as a modification cascade: first t6A at pos. 37, then addition of mcm5U at pos. 34, followed by conversion to mcm5s2U at pos. 34, and finally conversion to ms2t6A at pos. 37; see Fig. 6 for annotated spectra. Based on our data alone it is impossible to determine whether these and other cases correspond to partial modifications of a particular tRNA, or to mixtures of differently modified tRNAs from separate genes. However, overall these results support newer findings that question the stoichiometric and static nature of tRNA modifications, and favor the notion of a complex and dynamic tRNA modification landscape[38].

## Discussion

NASE is an open-source database search engine for RNA, optimized for high-resolution MS data. It supports arbitrary modifications, salt adducts, and FDR estimation through a target/decoy search strategy. Moreover, integration with the OpenMS toolbox enables high-quality data visualization, e.g. for manual validation of spectral assignments, and label-free quantification of RNA oligonucleotides. We have tested NASE against a range of sample types and complexities, spanning synthetic nucleic acids, in vitro-transcribed sequences, and cell extracts. In all of these experiments we have been able to effectively identify RNA sequences and their modifications.

NASE contains many unique functionalities that are not currently realized in other database search tools for RNA. To our knowledge, no other tools account for precursor mass defect resulting from instrumental selection of higher isotopologue peaks. This functionality is a major contributor to the excellent performance of NASE in identifying longer oligonucleotides compared to other database-matching tools. NASE also provides powerful correction for cation adduction events, which lessens the impact of sodium and potassium ions on sequence characterization. In addition, OpenMS in general and NASE

specifically were designed to be fast. Our search times for complex samples are orders of magnitude faster than other tools. The searches on the NME1 and let-7 data take seconds; the much more complicated 23-modifications searches of the tRNA dataset took <30 min per file on our server (using 40 parallel threads). For comparison, an analogous search using RNAModMapper was not feasible, with an estimated running time of one month. An equivalent search with Ariadne did not return any modified oligonucleotides.

The open-source nature of OpenMS and NASE enables users to modify the software to fit their specific needs, to extend the existing functionality, and to create new interoperating programs. Already, many analysis tools have been implemented within the OpenMS framework to support mass spectrometry-based proteomics and metabolomics experiments. The present work, and here in particular the pioneering application of label-free quantification, gives a foretaste of the power of leveraging these methods for the analysis of nucleic acid data. Future developments will streamline the use of OpenMS tools and algorithms, e.g. for improved quantification and comparisons across many samples, in the field of epitranscriptomics. In conclusion, the development of NASE is an important step towards the large-scale analysis of RNA by mass spectrometry.

## Methods

**Liquid chromatography-tandem mass spectrometry**. RNA samples were separated by reversed-phase ion-pair liquid chromatography (using 200 mM HFIP + 8.5 mM TEA in $H_2O$ as eluent A, and 100 mM HFIP + 4.25 mM TEA in methanol as eluent B) and characterized by negative ion MS/MS in a hybrid quadrupole-orbitrap mass spectrometer (Q Exactive HF, Thermo Fisher). A gradient of 2.5 to 25% eluent B eluted oligonucleotides from various lengths of nanoflow Acclaim PepMap C18 solid phase (Thermo Fisher) at 200 nL per minute. The length of the gradient was varied according to the complexity of the sample. Precursor ion spectra were collected at a scan range of 600–3500 m/z at 120k resolution in data-dependent mode, with the top five MS1 species selected for fragmentation and MS2 at 60k resolution.

**RNA samples**. A variety of RNA samples were characterized by nanoflow LC-MS/MS (nLC-MS/MS) and sequence analysis performed using NASE. Initial work was carried out on a mature *Drosophila* let-7 sequence that was prepared by solid-phase synthesis and purchased from IDT. This sequence is a 21 nt long microRNA that was among the first miRNAs to be characterized[39]. The RNA was chemically synthesized in unmethylated and methylated forms, i.e. with or without a 2′-O-methyluridine (Um) at position 21. A sample was prepared by mixing both forms, and was characterized by nLC-MS/MS without further processing, but with varying normalized collision energy (NCE) settings to give different levels of precursor fragmentation.

Subsequent experiments were carried out on NME1, a 340 nt long *Saccharomyces* lncRNA. NME1 RNA was generated by in vitro transcription, and two samples with and without NCL1 enzyme treatment were prepared. NCL1 is a yeast RNA methyltransferase that catalyzes the 5-methylcytidine (m5C) modification[40]. RNA was extracted and digested with RNase T1; this endonuclease produces shorter oligonucleotides by cleaving immediately after guanosine residues. nLC-MS/MS analysis of technical triplicates of 100 ng of oligonucleotides was performed.

For the human ribosomal RNA dataset, total RNA was extracted from HAP1 tissue culture by using Qiazol reagent according to the manufacturer's instructions. Samples were generated by size-exclusion chromatography of total RNA.[41] Briefly, total RNA was fractionated through an Agilent Bio SEC-5 column using a mobile phase of 100 mM ammonium acetate (pH 5) at a flow rate of 250 uL per minute. Two fractions containing long ribosomal (18S and 28S) RNA were digested with RNase T1 and 250 ng of oligonucleotides were subsequently analyzed by nLC-MS/MS in technical triplicate.

The most complex sample was a solution of digested crude human cellular tRNA, which was isolated in three biological replicates from HAP1 tissue culture using an RNeasy kit (Qiagen) according to the manufacturer's instructions. Briefly, RNAs can be fractionated by length by differential elution, with RNAs less than 200 nucleotides mostly made up of tRNA, and the larger fraction being mostly rRNA. The "shorter" RNA fraction was digested with RNase T1, and the resultant oligonucleotides were characterized by nLC-MS/MS in technical triplicate, with 100 ng being injected each time.

**NASE implementation**. NASE was implemented in C++ within the OpenMS framework. The OpenMS library was extended with classes representing (modified)

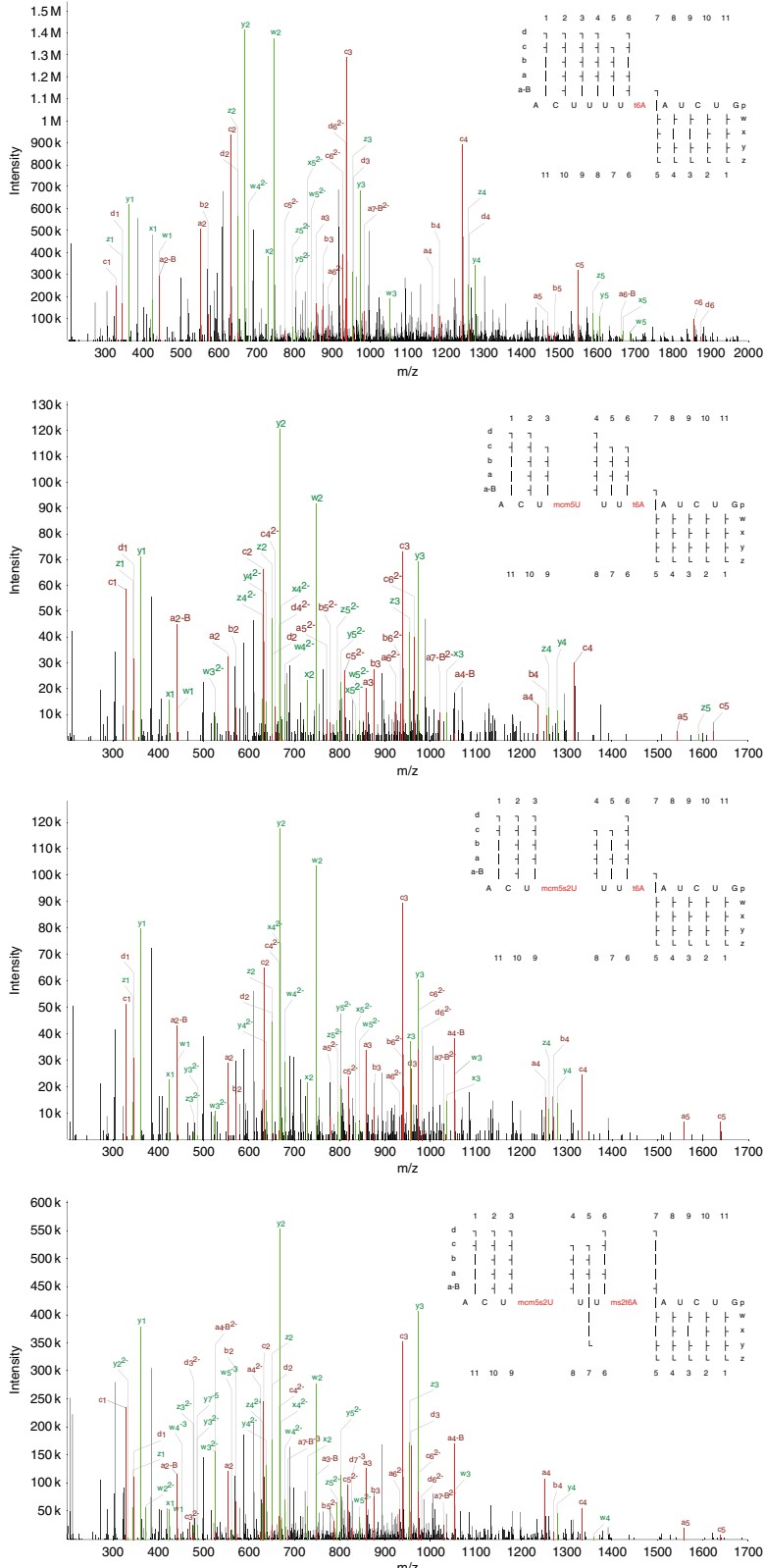

**Fig. 6 Annotated spectra of four oligonucleotides originating from tRNA-Lys^TTT.** Matched a-B/a/b/c/d ions are shown in red, w/x/y/z ions in green. All oligonucleotides cover the same region around the anticodon, but represent different modification states, evocative of stepwise addition and conversion of modifications by the associated enzymes. Each of these spectra had a precursor with a charge state of −5 with the exception of "ACUUUU(t6A)AUCUG", which was −3. All of these spectral assignments had a hyperscore of at least 140 in NASE.

**Table 2 Parameters for NASE searches of different datasets.**

| Parameter | Synthetic miRNA | NME1 | Human rRNA | Human tRNA |
|---|---|---|---|---|
| Fragment ions | a-B, a, b, c, d, w, x, y, z | a-B, a, b, c, d, w, x, y, z | a-B, a, b, c, d, w, x, y, z | a-B, a, b, c, d, w, x, y, z |
| Precursor mass tol. | 3 ppm | 3 ppm | 3 ppm | 3 ppm |
| Fragment mass tol. | 3 ppm | 3 ppm | 3 ppm | 3 ppm |
| Adducts | $Na^+$, $K^+$, $Na_2^{2+}$, $K_2^{2+}$, $NaK^{2+}$, $Na_3^{3+}$, $K_3^{3+}$, $Na_2K^{3+}$, $NaK_2^{3+}$ | $Na^+$, $K^+$ | $Na^+$, $K^+$, $NaK^{2+}$ | $Na^+$, $K^+$ |
| Modifications | Um (fixed) | m5C | mA?, mC?, mG?, mU?, m6,6A, m1acp3Y, ac4C | See Table 1. |
| Mods. per oligo. | – | 2 | 3 | 3 |
| Precursor mass offsets | 0–4 | −1 to 3 | −1 to 3 | 0–4 |
| Digestion | Unspecific cleavage | RNase T1 | RNase T1 | RNase T1 |
| Missed cleavages | – | 1 | 2 | 2 |

ribonucleotides (based on data from the MODOMICS database[42]), RNA sequences, and endoribonucleases. A generalized data structure for spectrum identification results (supporting peptides/proteins, nucleic acid sequences, and small molecules) and an algorithm for theoretical spectrum generation of RNAs were added as well. NASE itself is an executable tool that supports the common interface of The OpenMS Proteomics Pipeline[26].

Data processing with NASE works as follows: Inputs are an RNA sequence database (FASTA format) and a mass spectrometry data file (mzML format). RNA sequences are digested in silico using enzyme-specific cleavage rules for the user-specified RNase. Tandem mass spectra are pre-processed (intensity filtering, deisotoping) and mapped to oligonucleotides based on precursor masses. Mass offsets due to salt adducts or precursor selection from heavier isotopologue peaks can be taken into account. Next, theoretical spectra of relevant oligonucleotides in the appropriate charge states are generated and compared to the experimental spectra; matches are scored using a variant of the hyperscore algorithm[43]. If the sequence database contains decoy entries, the resulting oligonucleotide-spectrum matches can be statistically validated through the automatic calculation of $q$-values, a measure of the FDR[44]. Supported output formats are an mzTab-like[45] text file, suitable for further analysis, and an XML file, suitable for visualization in TOPPView.

In order to provide support for label-free quantification of identified oligonucleotides, NASE interfaces with the OpenMS tool FeatureFinderMetaboIdent (FFMetId). FFMetId handles the core step of the quantitative workflow, the detection of chromatographic features in the LC-MS data. As a variant of the proteomics tool FeatureFinderIdentification[46], FFMetId provides targeted feature detection for arbitrary chemical compounds. NASE can write an output file with all relevant information about the oligonucleotides it identified, which is directly suitable as an input file for FFMetId.

**Sequence database searches**. For NASE analyses, all proprietary raw files were converted to mzML format[47] without compression and with vendor peak-picking using MSConvert[48] (https://github.com/ProteoWizard). The full list of fragment ion types (a-B, a, b, c, d, w, x, y, z) was considered for peak matching. Precursor and fragment mass tolerance were both set to 3 ppm.

For the synthetic let-7 data, an extensive set of potential adducts ($Na^+$, $K^+$, $Na_2^{2+}$, $K_2^{2+}$, $NaK^{2+}$, $Na_3^{3+}$, $K_3^{3+}$, $Na_2K^{3+}$, $NaK_2^{3+}$) was used during the search because of the substantial salt that remained from the RNA synthesis reactions. Two copies of the let-7 sequence, one with a fixed 2′-O-methylation of uridine (Um) at the 5′ position, were specified in the FASTA sequence file. The small size of the sequence database prevented the use of a target/decoy approach for FDR estimation. We thus used a stringent hyperscore cutoff of 150 (corresponding to the 1% FDR in the tRNA sample, see below) to define a high-confidence set of results.

In the three other datasets, results from target/decoy database searches were initially exported from NASE at 10% FDR (spectrum match-level), then further filtered in post-processing depending on the analysis.

For the NME1 data, the sequence database contained the NME1 (target) sequence as well as a shuffled decoy sequence.

For the human rRNA data, to allow direct comparison with published results, we used a sequence database containing 18S and 28S rRNA from TK6 cells (plus reversed decoys). A set of seven variable modifications, including monomethylations of all four canonical ribonucleotides, was defined based on known ribosomal PTMs.

In our search of the tRNA data, 23 variable modifications (based on reported modifications in human tRNA[36]) were specified, at a maximum of three modifications per oligonucleotide. The FASTA file contained 420 human tRNA sequences collected from the tRNA sequence database tRNAdb[35] (http://trna. bioinf.uni-leipzig.de) plus the same number of reversed decoy sequences.

See Table 2 for additional dataset-specific parameters.

**Search engine comparison**. The NME1 data was processed with two other publicly available RNA identification engines, in addition to NASE: Ariadne[21] and RNAModMapper[23]. To this end, the raw files were converted to MGF format using MSConvert. Cleavage and variable modification settings in the searches were the same as for NASE and appropriate for the samples.

For Ariadne, the online version at http://ariadne.riken.jp was used in July 2019. The "Calc as partial modifications" option was enabled. The precursor and fragment mass tolerances were left at their default values (5 and 20 ppm). Alternatively, using the parameters from the Taoka et al.[49] (20 and 50 ppm) made no appreciable difference for Ariadne's performance in our tests.

For RNAModMapper, a program version from July 2018 was used with settings recommended by the author, Ningxi Yu. All available ion types (a-B, w, c, y) were enabled; precursor and fragment mass tolerances were set to 0.02 and 0.1 Da, respectively.

**Label-free quantification**. In order to perform label-free quantification on the NME1 dataset, target coordinates (chemical sum formulas, charge states, median retention times) for oligonucleotides identified at 1% FDR were exported from NASE. Based on these coordinates, feature detection in the LC-MS raw data (mzML files) was carried out with the OpenMS tool FeatureFinderMetaboIdent. The results were exported to text format using OpenMS' TextExporter, for subsequent processing and visualization in R 3.5.1[50]. Results from both NME1 samples were merged and feature intensities for oligonucleotides were summed up over multiple charge and adduct states, where available. To equalize differences between replicates and ensure comparability between conditions, the quantities for oligonucleotides that differed only in the localization of a cytidine methylation, as well as for the overlapping oligonucleotides "UAACCCAAUGp" and "UCA-CAAAUCGp" (and their variants), were aggregated.

**Reporting summary**. Further information on research design is available in the Nature Research Reporting Summary linked to this article.

## Data availability

Mass spectrometry data files and search results (as well as label-free quantification results for the "NME1" dataset) were deposited in the PRIDE[51] repository with dataset identifiers PXD012094 [https://www.ebi.ac.uk/pride/archive/projects/PXD012094] (synthetic let-7), PXD016308 [https://www.ebi.ac.uk/pride/archive/projects/PXD016308] (NME1), PXD016323 [https://www.ebi.ac.uk/pride/archive/projects/PXD016323] (human rRNA) and PXD016328 [https://www.ebi.ac.uk/pride/archive/projects/PXD016328] (human tRNA).

## Code availability

Source code for OpenMS, including NASE, is available on GitHub (https://github.com/OpenMS/OpenMS) under a three-clause BSD license. R scripts for post-processing of NASE results are available by request.

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

## Acknowledgements

We would like to thank the following people: Lina Vasiliauskaitė and Alan Hendrick for preparing the HAP1 tRNA and rRNA samples, respectively. Jack Rogan for performing the RNA fractionation during rRNA sample generation. Ningxi Yu for suggesting optimal parameters for running his RNAModMapper tool. Hiroshi Nakayama for sharing the TK6 rRNA sequences used by Taoka et al.[32] All contributors to OpenMS; especially Hannes Röst for his efforts and useful feedback during the code review for this project. B.A.G. acknowledges funding from NIH grant GM110174 and a UPenn Epigenetics Institute Pilot grant.

## Author contributions

S.W., T.S., and H.W. developed the software. S.W., B.A., and H.W. analyzed the data and wrote the paper. B.A. performed the LC-MS/MS experiments. H.S-R. prepared the NME1 samples. O.K., T.K., and B.A.G. provided resources and high-level supervision. All authors read and approved the manuscript.

## Competing interests

T.K. is a founder and director of STORM Therapeutics Limited, Cambridge, UK. B.A. and H.W. are full-time employees of STORM Therapeutics Limited, Cambridge, UK. S. W., T.S., H.S-R., O.K. and B.A.G. declare no competing interests.
