## [Peer Review File · Nature Communications]

Reviewers' comments:

Reviewer #1 (Remarks to the Author):

The paper describes a new software pipeline for analysis of RNA sequence modifications. This is a relatively novel and growing area, for which there is currently not much software, and little support for statistical analysis of data. The manuscript is well written and clear. I tried out the software on a Windows platform and it ran without problems on the test data sets. Overall I am very supportive of publication, and I did not spot any minor issues in need of correction

Reviewer #2 (Remarks to the Author):

The current manuscript describes a new software for the automatic interpretation of RNA tandem MS spectra with the main focus on identifying RNA modifications. An additional feature is an option for label-free quantitation based on comparison of LC-MS/MS runs.

A series of attractive features are present in the software compared to "competitors", e.g. calculation of false discovery rates and isotopic correction of precursor ion mass. However, the software's ability to identify RNA modifications is by no means convincingly demonstrated, neither with the in vitro enzyme-modified setup nor the human tRNA.

For the in vitro modified setup, seven out of ten modified oligonucleotides are present at equal intensity in both the modified half and the unmodified control half (figure 3b). The explanation about chromatographic carry-over does not hold – no normal chromatographic setup leaves 50% of an analyte on the column to be detected in the subsequent run. Hence, this experiment had de facto reported 70% false positives if there was not an unmodified control to cause alert; and there will not be an unmodified control in most modification studies! As a consequence of "modified" oligonucleotides being detected in the unmodified sample, the claimed quantitation option is also dubious. The authors use experimental artifacts to substantiate the potential for quantitation! In the human tRNA sample, tRNA-Val(ACC) 3-1 is used to demonstrate the ability to locate modifications and it is stated that the tRNA database (tRNADB) is the source of info on previously identified modifications. I am, however, unable to find this information in tRNADB. I find a single different tRNA-Val isotype with a quite different modification pattern – will the authors please identify the source more precisely? Even given that the mistake is on my side, the authors state that four out of their eight identified modifications are not in agreement with the tRNADB sequence (legend to figure 5a), which again raises suspicion about the validity of modification identification and assignment.

The above inability to identify modifications with an acceptable certainty suggests that the software fails in the peak selection for the interpretation of tandem MS data. This notion finds support in the often very low-intensity ions that assigned in the spectra (= are considered relevant for spectra interpretation). Another problem with the in vitro modified sample and the tRNA sample is that the expected level and pattern of modification is not really known, so the samples are not appropriate for testing the efficiency of the software. I suggest that the testing is done on better characterized samples such as ribosomal RNA from a model organism, where appropriate controls, consisting of rRNA from mutants deficient in selected modification, can easily be obtained.

A series of minor points, of which some are more comments than critique:

- Line 18: I dislike the term "chemical modification", which suggests modifications being induced by an unspecific chemical reaction. The modifications are made by (highly) specific enzyme systems.
- Line 44-66: The authors stretch the justification of MS approaches. For example, is the 13% error rate/read in single molecule sequencing cancelled out by averaging multiple reads; and the MS analysis in sequence context gives little information on exact nature of modifications compared to e.g. antibody/NGS approaches.

- Line 105: Are there any expectation regarding NCL1 methylation pattern? Would be useful for the subsequent evaluation of modification ID data.
- Line 143-144: Not all sequence ions are equally likely (see e.g. PMID 18799321). Has this been considered in the software (and if not, to what extent would it be relevant)? This point also pertains to the problems in modification ID.
- Figure 1: Several of the intense peaks are not assigned, whereas some of the assigned sequence ions are hardly visible. The authors should elaborate on this issue. Does this contribute to the problems with modification placing?

Much of the text on the figure is too small.

- Line 180. Are 2'-3' cyclic phosphate products observed? Maybe these "intermediate" RNase T1 products should be a search parameter in the software like the salt adducts?
- Line 220 (approx.): has the impact of G-nucleotide modification of RNase T1 cleavage been taken into account? E.g. ribose methylation completely blocks enzymatic cleavage.
- Line 230-231: Partial cleavage products should be observed if some parts of tRNA are less RNaseT1 accesible. Is this the case (the software allows this option in the search)?
- Table 1: How are "unique oligonucleotides" defined?
- Line 288-289: It would be nice with more information on how the correction for "wrong isotope assignment" works.

Reviewer #1 (Remarks to the Author):

The paper describes a new software pipeline for analysis of RNA sequence modifications. This is a relatively novel and growing area, for which there is currently not much software, and little support for statistical analysis of data. The manuscript is well written and clear. I tried out the software on a Windows platform and it ran without problems on the test data sets. Overall I am very supportive of publication, and I did not spot any minor issues in need of correction

We would like to thank this reviewer for this very supportive review of our work.

Reviewer #2 (Remarks to the Author):

The current manuscript describes a new software for the automatic interpretation of RNA tandem MS spectra with the main focus on identifying RNA modifications. An additional feature is an option for label-free quantitation based on comparison of LC-MS/MS runs.

A series of attractive features are present in the software compared to “competitors”, e.g. calculation of false discovery rates and isotopic correction of precursor ion mass. However, the software’s ability to identify RNA modifications is by no means convincingly demonstrated, neither with the in vitro enzyme-modified setup nor the human tRNA.

Validation in this field is difficult in the absence of clearly defined benchmark samples. For our resubmission we have reacquired and reanalyzed two of our datasets. We have also conducted additional experiments on rRNA and compared the results from our software to published data. We believe that the concurrence of our results with well-validated data convincingly demonstrates our software’s ability to identify RNA modifications. We hope that the reviewer shares our assessment, as we agree with his inference that the modification pattern of rRNA is the closest the field has to a reference sample.

For the in vitro modified setup, seven out of ten modified oligonucleotides are present at equal intensity in both the modified half and the unmodified control half (figure 3b). The explanation about chromatographic carry-over does not hold – no normal chromatographic setup leaves 50% of an analyte on the column to be detected in the subsequent run. Hence, this experiment had de facto reported 70% false positives if there was not an unmodified control to cause alert; and there will not be an unmodified control in most modification studies! As a consequence of “modified” oligonucleotides being detected in the unmodified sample, the claimed quantitation option is also dubious. The authors use experimental artifacts to substantiate the potential for quantitation!

We have reacquired data from the NME1 samples in multiple replicates (using an improved chromatographic gradient on a fresh column), analysed them with the newest version of NASE, and applied stricter filtering to exclude unreliable identifications. The results are clearer now and our manuscript has been updated accordingly.

(While no longer in the manuscript, we can shed some light on the previous results: There are two different factors that can together explain the somewhat contradictory observations. Carry-over is one - that can explain the residual signal for the methylated oligonucleotide in Sup. Fig. 3. However, we agree and thank

the reviewer for pointing out that it is not the relevant explanation for methylated oligonucleotides that were quantified at a constant level in both samples. The second, and more relevant, factor is misidentification of spectra. As we mentioned in the text, "single-hit" identifications are generally considered unreliable - we included them for completeness in this evaluation, but that may have been unhelpful, as the reviewer considered the hits identified with a single spectral match to be the same as those that were identified with many. We have filtered out these "single-hit" identifications in our updated analysis to aid interpretation.)

In the human tRNA sample, tRNA-Val(ACC) 3-1 is used to demonstrate the ability to locate modifications and it is stated that the tRNA database (tRNAdb) is the source of info on previously identified modifications. I am, however, unable to find this information in tRNAdb. I find a single different tRNA-Val isotype with a quite different modification pattern – will the authors please identify the source more precisely? Even given that the mistake is on my side, the authors state that four out of their eight identified modifications are not in agreement with the tRNAdb sequence (legend to figure 5a), which again raises suspicion about the validity of modification identification and assignment.

We have generated a much more extensive tRNA dataset (three biological and three technical replicates), searched it with NASE using an updated set of variable modifications, and analyzed the results more stringently. Information on known tRNA modifications is now taken from the newer, slightly more comprehensive data in MODOMICS. However, given the incompleteness of this data and the ambiguities in attributing identified oligonucleotides to tRNA genes, we are now also more cautious about making comparative statements in the paper.

Our main reason for including the tRNA dataset is not to validate our software's ability to identify modifications - we hope to have achieved this based on the other datasets. Primarily, we aim to show that the analysis of such complex samples, involving a previously unthinkable number of variable modifications, is now feasible and can generate biologically meaningful results.

The above inability to identify modifications with an acceptable certainty suggests that the software fails in the peak selection for the interpretation of tandem MS data. This notion finds support in the often very low-intensity ions that assigned in the spectra (= are considered relevant for spectra interpretation).

We believe this critical view comes more from our - perhaps not optimal - presentation of the results in the previous manuscript than from true shortcomings of our software. NASE is a new tool that will mature with increased use and continued development, and builds on key developments in the proteomics field over the last decades, such as the use of decoy databases and label-free quantification. The results that it generates are imperfect (as with any software), but are already much more comprehensive and reliable than results of alternative tools. We feel that the revised version of our manuscript now convincingly shows the utility of RNA mass spectrometry supported by NASE.

Low-intensity ions can be assigned to theoretical fragments by NASE, but their contribution to the score of an oligonucleotide-spectrum match is partially based on their (relative) intensity, i.e. low. Users who prefer to exclude peaks below a certain intensity threshold can easily achieve this by filtering their data prior to NASE analysis. Future tests will have to show whether there is an optimal way of filtering that can reliably improve NASE analysis results, but we consider this beyond the scope of the current manuscript.

Another problem with the in vitro modified sample and the tRNA sample is that the expected level and pattern of modification is not really known, so the samples are not appropriate for testing the efficiency of the software.

I suggest that the testing is done on better characterized samples such as ribosomal RNA from a model organism, where appropriate controls, consisting of rRNA from mutants deficient in selected modification, can easily be obtained.

As stated earlier in this response, we agree that ribosomal RNA would have the most well-characterised post-transcriptional modifications of any biological sample. As a result, we have now included new experimental data collected from rRNA, and have compared the identity and site of modification derived from our software with published data. Our software gives excellent agreement with the current state of the art (84% exact identification type and site) and massively reduces the need for manual assessment of mass spectra.

A series of minor points, of which some are more comments that critique:

- Line 18: I dislike the term “chemical modification”, which suggests modifications being induced by an unspecific chemical reaction. The modifications are made by (highly) specific enzyme systems.

We have replaced this with the term “post-transcriptional modification” in the manuscript.

- Line 44-66: The authors stretches the justification of MS approaches. For example, is the 13% error rate/read in single molecule sequencing cancelled out by averaging multiple reads; and the MS analysis in sequence context gives little information on exact nature of modifications compared to e.g. antibody/NGS approaches.

We have removed the “13% error rate” as a point of contention, but believe that our overall view is justified. It is true that consensus approaches can be used to reduce error rates, but the same is true for MS-based methods, where information from multiple spectra can be aggregated. We would respectfully disagree that antibodies give specific information, given the current controversy in the field about how to interpret antibody-based approaches (the interpretations of very similar NGS datasets give wildly different estimates of m1A prevalence, for example). In addition, antibodies do not exist for the majority of modifications, and in the worst case can lack specificity in a variety of analyses. For example, most of the commercial antibodies against m5C do not bind avidly, and antibodies raised against m6A also recognise m6Am - these modifications are easily resolvable by mass spectrometry.

In general, many modifications are uniquely identified by their mass shifts in MS-based methods. For others, extensions of the tandem MS approach discussed here can be applied to gain further insights. For example, multiple rounds of fragmentation (MS3) can generate isomer-specific fragments that allow discrimination between positional isomers, e.g. of monomethylations. (Poster presented by Hiroshi Nakayama at ASMS 2019: “Mass spectrometry-based identification of mono-methylated RNA nucleoside positional isomers: Application for structural analysis of RNA modifications in the Leishmania ribosome”)

- Line 105: Are there any expectation regarding NCL1 methylation pattern? Would be useful for the subsequent evaluation of modification ID data.

Unfortunately no such data is available. The interaction between NCL1 and NME1 has been established within the Kouzarides lab (unpublished data), but modification sites have not been determined.

We considered the use of bisulfite treatment with a Sanger sequencing readout of allelic variation, but concluded that the results would not be sensitive enough for comparison without extensive optimisation, which seems an unreasonable undertaking given the scope of the work described within our manuscript.

- Line 143-144: Not all sequence ions are equally likely (see e.g. PMID 18799321). Has this been considered in the software (and if not, to what extent would it be relevant)? This point also pertains to the problems in modification ID.

This has been considered and the software already contains options for ion type-specific weighting in the scoring function. However, weighting is not used at the moment and is not exposed to the user. Since relative ion intensities may potentially depend on several factors (instrument/mass analyzer, charge state, collision energy...) we did not feel comfortable prescribing specific weight settings at this point. An analysis of relative ion intensities and their impact on scoring is planned as a future extension to this work.

- Figure 1: Several of the intense peaks are not assigned, whereas some of the assigned sequence ions are hardly visible. The authors should elaborate on this issue. Does this contribute to the problems with modification placing?

Much of the text on the figure is too small.

The intensity of the matched peaks is considered in the scoring function already, i.e. low-intensity peaks contribute less than high-intensity ones. Using OpenMS tools it is possible to pre-filter the spectra to remove low-intensity peaks before database searching, but we have not investigated the effect of including this additional step in the present series of experiments.

The current generation of NASE does not try to assign neutral losses from the RNA, such as depurination or water loss. We think it is likely that these account for high-intensity unassigned peaks, given the current models of gas-phase RNA fragmentation. In addition, we do not try to assign internal fragmentation events (those that do not include each terminal of the oligonucleotide), as these are not as informative for deriving the sequence. Pre-filtering of spectra, support for neutral losses and other improvements to the scoring function are all items that we are considering for future versions of the software.

We have increased the text size on the figure, and hope that it is now easier to interpret.

- Line 180. Are 2'-3' cyclic phosphate products observed? Maybe these "intermediate" RNase T1 products should be a search parameter in the software like the salt adducts?

We do not routinely observe 2',3'-cyclic phosphates at the MS1 level. Our understanding is that these will be hydrolyzed to 3' phosphates in aqueous solutions like the buffers used for LC-MS/MS analysis.

- Line 220 (approx.): has the impact of G-nucleotide modification of RNase T1 cleavage been taken into account? E.g. ribose methylation completely blocks enzymatic cleavage.

This is an excellent question, and thanks to the reviewer for raising it. We are aware of this issue; the interplay between RNase T1 cleavage and 2'-O-methylation of guanosine is not properly supported and is a current limitation in our software. Adding support is non-trivial, but is intended for the future. The problem is mitigated by support for missed cleavages and by the introduction of generic, position-unspecific modification codes for methylations (e.g. "mG" instead of "m1G" or "Gm") in a recent version of NASE.

- Line 230-231: Partial cleavage products should be observed if some parts of tRNA are less RNaseT1 accessible. Is this the case (the software allows this option in the search)?

It is rare to see oligonucleotides with missed cleavages in our tRNA results, and certainly not often enough to draw a meaningful conclusion. Usually, there is either a guanosine modification, or a large modification nearby that may inhibit the endonuclease access to the site. In fact, only 13 different oligonucleotides (10 when modifications are ignored) with missed cleavages at unmodified guanosines were detected.

Conversely, cleavages at unexpected sites (i.e. not after G) were not considered in our database searches. “Unspecific cleavage” is supported as a parameter in NASE, but would incur too much computational overhead in a dataset as complex (heavily modified) as tRNA.

- Table 1: How are “unique oligonucleotides” defined?

Oligonucleotides with different, discrete sequences or modification patterns. The table has been simplified and this information is no longer included.

- Line 288-289: It would be nice with more information on how the correction for “wrong isotope assignment” works.

There is now a more detailed explanation in the section “A powerful new search engine for RNA MS data”.

REVIEWERS' COMMENTS:

Reviewer #2 (Remarks to the Author):

I thank the authors for a thorough response to the critique raise during the first submission. The revised manuscript (and the comments) constructively address all my previous points.